# Effect of Side Blowing on Fluid Flow and Mixing Phenomenon in Gas-Stirred Ladle

Rong Cheng [1], Liangjin Zhang [2], Yanbin Yin [1] and Jiongming Zhang [1,*]

1  State Key Laboratory of Advanced Metallurgy, University of Science and Technology Beijing,
   Xueyuan Road 30, Beijing 100083, China; b20160497@xs.ustb.edu.cn (R.C.); ustbyby@ustb.edu.cn (Y.Y.)
2  China National Construction Steel Quality Supervision and Test Centre, Central Research Institute of Building
   and Construction Co., Ltd., Mcc Group, Xitucheng Road 33, Beijing 100088, China; zy188cmcc@163.com
*  Correspondence: jmz2203@sina.com; Tel.: +86-010-82376597

**Abstract:** To investigate the gas agitation characteristics of side blowing, the fluid flow and mixing phenomenon in a 1:3 scale model ladle of a 150 t industrial gas-stirred ladle with bottom and side plugs were studied by using physical and numerical modelings together. Side blowing enhanced the horizontal flow of water in the model ladle. Compared with bottom blowing, side blowing that is close to the ladle bottom with more than two plugs increases the average velocity of water, which represents the agitation power, improves the uniformity of water velocity distribution, reduces the stagnant region rate, and shortens the mixing time. The mixing time of dual bottom plugs is almost 1.5 times of that of four side plugs at 116 mm under the same flow rate. The mixing time is not only influenced by the agitation power but also by the uniformity of water velocity distribution. Although the agitation power of four side plugs at 450 mm under the flow rate of 1.8 $m^3$/h is about 1.5 times of that at 116 mm with 0.6 $m^3$/h. The mixing time of the 1.8 $m^3$/h flow rate is about 1.2 times of that of the 0.6 $m^3$/h because of the different water velocity distributions.

**Keywords:** gas-stirred; side blowing; mixing time; average velocity; stagnant region





## 1. Introduction

The alloying process is an important target during the secondary refining process. An alloying element could eliminate deleterious effects of other elements in molten steel and improve the physical and mechanical properties of steel products. Argon gas agitation is fully responsible for improving the fluidity of molten steel during ladle metallurgy. The chemical reactions [1,2], inclusion removal [3–5], and homogenization of the alloy and temperature [6–8] are closely related to the gas agitation in the ladle.

A large number of studies show that the mixing time has commonly been used to represent the gas agitation power in gas-stirred ladles [9–16]. In studies of Nakanishi [9] and Sinha [10], the relation between mixing time and agitation power were investigated by using a solution of 75% $H_2SO_4$ as a tracer in a water model ladle and derived as follows: $\tau_m = 800\varepsilon_m^{-0.4}$ [9] and $\tau_m = 692\varepsilon_m^{-0.89}$ [10] (where $\tau_m$ and $\varepsilon_m$ correspond to the mixing time and agitation power, respectively). Some researchers [11–14] correlated the mixing time as simple functions of agitation power, liquid depth, and radius of the ladle with the experiment of tracer homogenization in a water modeling ladle. In the work of Mazumdar et al. [13], the mixing time was defined as the time required for the monitoring point concentration to fall continuously within a 5% deviation of the well mixed/homogeneous value, and an empirical expression of mixing time was derived: $\tau_m = 37\varepsilon_m^{-0.33}R^{1.66}L^{-1.0}$ (where $L$ and $R$ correspond to the liquid depth and ladle radius, respectively). Zhu [15] and Amaro-Villeda [16] summarized mixing time correlations reported in the literature under different experimental conditions, such as gas flow rates, number of nozzles, nozzle radial position, and thickness of oil.

Physical modeling [9–30] was usually adopted to study the hydrodynamics and mixing phenomena in gas-stirred ladles with bottom plugs. The probe measurements of pH and conductivity were widely used to measure the mixing time of the tracer in a model ladle [9–25]. Some researchers [10,15,17] recognized that the mixing time is influenced by the addition position of the tracer and the probe positions in water modeling. It was reported that a ratio in the range of 1–1.5 for the height to the bottom diameter of the ladle is beneficial to the gas agitation in the ladle with bottom plugs [11,13,18]. The high flow rate contributed to reducing the mixing time in the ladle with bottom plugs [19,20]. Some studies [17,21–23] discussed the influences of plugs' configurations, such as plug number, plug radial position, and separation angle of dual plugs on the mixing time. Gómez et al. [23] recognized that a separation angle of 60° provides the best mixing efficiency. Tang et al. [24] found that the mixing time of every plug with different flowrates is shorter than that of every plug with same flowrate in a model ladle with dual bottom plugs. It has been reported [16] that the top slag layer increases the mixing time. It may partly be because the overlying slag dissipates a part of the input energy, which causes the velocity of liquid recirculation and the level of turbulence decrease significantly [25]. The energy dissipation rate for surface waves was less than 1%, and the energy dissipation rate for the pure liquid zone, plume zone, and spout was 36%, 22%, and 41%, respectively [26].

Probe measurements provide only local concentration evolution, and the solid probes may affect the fluid flow in small model ladle. Planar laser-induced fluorescence (PLIF) is a nonintrusive technique and provides a detailed contour of concentration in an entire plane of the model ladle [27,28]. The PLIF technique was implemented to measure mixing time in a 1/17 water model of a 200-ton ladle furnace by Jardón-Pérez et al. [27], who discussed the feasibility of the PLIF technique used to measure mixing time in physical modeling. The PLIF technique was experimentally implemented to measure temperature fields in a longitudinal plane of the gas-stirred ladle model by Jardón-Pérez et al. [28]. Convective flow and turbulent diffusion were the two primary transport mechanisms of the mixing phenomenon [29]. Particle image velocimetry (PIV) technology [23,27,28,30–32], used to measure the flow patterns and turbulent intensity of fluid in physical modeling, was an effective tool to understand the effect of fluid flow on the mixing phenomenon. González-Bernal et al. [23] recognized that colorant and tracer dispersion techniques are complemented fairly well with PIV results, easily identifying areas of stagnation or recirculation within the model ladle. They found that the mixing time is significantly affected by the number, size, and location of the recirculation in the bulk of the fluid. The hydrodynamic performance was analyzed by Jardón-Pérez et al. [30] through physical modeling and PIV measurements. The analysis showed that large mean velocity and turbulent kinetic energy led to a short mixing time. In the previous work of Jardón-Pérez et al. [27], the turbulent kinetic energy contour plots obtained by PIV showed that the dead zone with small turbulent kinetic energy of eccentric injection is smaller than that of the centric injection. Therefore, the mixing time is lower for the eccentric injection than for the centric injection.

Numerical modeling [21,33–38] was widely used to investigate the fluid flow and mixing phenomena in industrial ladles. A numerical model of alloy homogenization in a ladle with bottom plugs was developed by Geng et al. [21] with the Euler–Euler approach to investigate the influences of gas flow rate and plug configurations on the mixing time. The mixing time was proportional to the value for the 0.2676 powers of gas flow in their study. Jauhiainen et al. [33] studied the effect of the addition position of alloy on the mixing time in a gas-stirred ladle with the Euler–Euler approach. Cloete et al. [34,35] conducted a mathematical model of alloy homogenization in a gas-stirred ladle with the Euler–Lagrange approach. They found that the mixing time was influenced by the number of plugs, plug positions, and the ratio of height to the diameter of the ladle. Ganguly et al. [36] discussed the effects of plug configurations and the addition position of alloy on the mixing time in a ladle with bottom plugs numerically. The effects of porous plug location, separation angle of two porous, and plugs gas flow rate on the fluid flow and

the mixing phenomena were studied by Duan et al. [37] using a Eulerian–Lagrangian approach. They recognized that the separation angle of 90° is recommended to improve the flow field and mixing phenomena. The flow and separation phases in a whirlpool were built by Stachnik and Jakubowski [38] using a three-phase VOF (volume of fluid) approach. Their research deems the VOF three-phase model suitable to predict sedimentation and accumulation of sediment.

The above studies mainly focused on bottom blowing in a gas-stirred ladle. However, the fluid flowing, which is driven by bubbles was mainly along the vertical direction in the ladle with bottom blowing. The fluidity of fluid at the bottom edge of the ladle with bottom blowing was poor [39,40]—the dead zone where the fluid fluidity delayed the mixing time [10,27]. Few studies reported side blowing in gas-stirred ladle. In addition, most previous studies [9–16] just reported the effect of agitation power on the mixing time. Few studies discussed the effects of the uniformity of fluid velocity distribution on the mixing time.

For this paper, the comparisons of fluid flow and mixing condition in model ladle between side blowing and bottom blowing were investigated by physical and numerical modeling together. The different working conditions of side blowing, such as flow rate, plug number, and installing height of plug were considered. The effects of fluid flow characteristics especially the fluid average velocity and the fluid velocity distribution on the mixing time were analyzed.

## 2. Physical Modeling

### 2.1. Experimetal Principle

The trajectory of the gas jet was calculated by the initial inertia force of the gas injecting into the liquid and the buoyancy received in the liquid in Themelis' study [41]. For the gas agitation scaling, the ratio of the inertial force and buoyancy force in the plume was considered to achieve a flow similarity with the modified Froude number $(Fr_m = \frac{\rho_{gas} u_0^2}{(\rho_{liquid} - \rho_{gas}) g d_0} \approx \frac{\rho_{gas} Q^2}{\rho_{liquid}^2 g d_0^5})$ [41]. A large number of studies [9–26] on fluid flow in a gas-stirred ladle were conducted with physical modeling based on the geometric similarity and flow similarity. In the studies, the molten steel was represented by water, and the argon gas was represented by the air or nitrogen gas. A physical model using geometric similarity and flow similarity was developed to study the molten steel flow in industrial trials in this study. In the physical modeling, argon and molten steel were represented by nitrogen (25 °C) and water (25 °C). The relation of flow rates between the prototype and physical modeling can be correlated in Equation (1), which is based on the flow similarity $(Fr_{m,model} = Fr_{m,prototype})$.

$$Q_{nitrogen} = \frac{\rho_{water} d_{0,model}^{2.5}}{\rho_{steel} d_{0,prototype}^{2.5}} \sqrt{\frac{\rho_{argon}}{\rho_{nitrogen}}} Q_{argon} \tag{1}$$

where $d_{0,model}$ and $d_{0,prototype}$ are the feature sizes of plug exits of the model and prototype, because of the geometric similarity, $d_{0,model}/d_{0,prototype} = 1/3$, $\rho_{nitrogen}$ is the density of nitrogen (1.25 kg/m$^3$), $\rho_{water}$ is the density of water (1000 kg/m$^3$), $\rho_{argon}$ is the density of argon (1.784 kg/m$^3$), $\rho_{steel}$ is the density of steel height (7020 kg/m$^3$), $Q_{nitrogen}$ is the nitrogen flow rate in the physical modeling (NL·min$^{-1}$), and $Q_{argon}$ is the effective flow rate in the prototype (L·min$^{-1}$).

The influences of pressures and temperatures on the scaling corrections were considered in previous works [17,42]. The temperature and pressure of liquid are taken into consideration by using the ideal gas law $pV = nRT$ and $p_{in} = p_0 + \rho_{liquid} g H_{liquid}$, respectively. Overall, the scaling criterion is derived as follows:

$$Q_{nitrogen} = \frac{\rho_{water} d_{0,model}^{2.5}}{\rho_{steel} d_{0,prototype}^{2.5}} \sqrt{\frac{\rho_{argon}}{\rho_{nitrogen}} \frac{T_{steel}(p_0 + \rho_{water} g H_{water})}{T_{water}(p_0 + \rho_{steel} g H_{steel})}} Q_{argon} \tag{2}$$

where $T_{water}$ is the water temperature (298 K), $T_{steel}$ is the molten steel temperature in the prototype (1873 K), $p_0$ is the normal pressure (101,325 Pa), $p_{in}$ is the bottom injection pressure (Pa), and $Q_{argon}$ is the flow rate under normal conditions in the prototype (NL min$^{-1}$), $H_{water}$ and $H_{steel}$ are the heights of filled water and filled molten steel in Table 1, respectively.

**Table 1.** Dimensional parameters of prototype and model.

| Parameters | Prototype | Model |
|---|---|---|
| Top diameter of ladle (m) | 3.138 | 1.046 |
| Bottom diameter of ladle (m) | 2.839 | 0.946 |
| Height of ladle (m) | 3.849 | 1.283 |
| Liquid depth H (m) | 3.348 | 1.116 |
| Taper of ladle (°) | 2.2 | 2.2 |
| Length of plug slots (mm) | 14 | 2.5 |
| Width of plug slots (mm) | 0.15 | 0.28 |
| Number of plug slots | 18 | 6 |

The flow rates of gas in model and prototype are shown in Table 2. In this paper, the flow rate of gas is the flow rate for a single plug.

**Table 2.** Flow rates of gas in model and prototype.

| Nitrogen (Nm$^3$/h) | 0.3 | 0.6 | 1.2 | 1.8 | 2.4 | 3.0 |
|---|---|---|---|---|---|---|
| Argon (NL/min) | 215 | 431 | 861 | 1292 | 1722 | 2153 |

## 2.2. Experimental Setup and Method

The geometrical dimensions of the model ladle were linked to that of a 150 t steel plant by using a ratio of 3. The ratio of plug ventilation area between the prototype and model is 9. The dimensional parameters of the prototype and model are shown in Table 1. Figure 1a shows the ladle and plugs of the model. Figure 1b shows the experimental apparatus. Five blowing modes, such as blowing by dual bottom plugs, dual side plugs at 116 mm, three side plugs at 116 mm, four side plugs at 116 mm, and four side plugs at 450 mm were studied in this paper. Figure 2 shows the three-dimensional coordinate system of the model ladle and the positions of the conductivity probes and plugs. The coordinates of the ladle bottom center point are (0, 0, 0). The positions of the conductivity probes referenced the positions of stagnant regions in Section 5. The horizontal and vertical angles between the side plugs and the ladle wall are 45° and 0°, respectively.

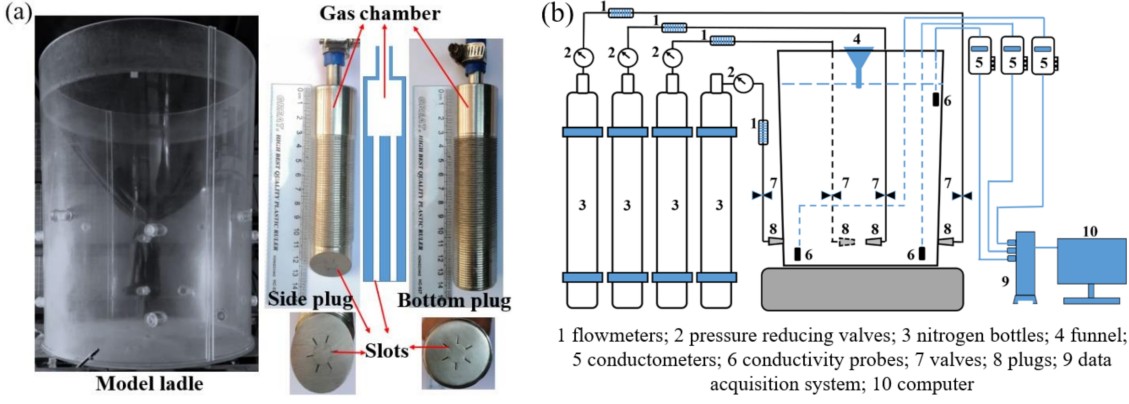

1 flowmeters; 2 pressure reducing valves; 3 nitrogen bottles; 4 funnel; 5 conductometers; 6 conductivity probes; 7 valves; 8 plugs; 9 data acquisition system; 10 computer

**Figure 1.** Experimental equipment: (**a**) ladle and plugs of model; (**b**) experimental apparatus.

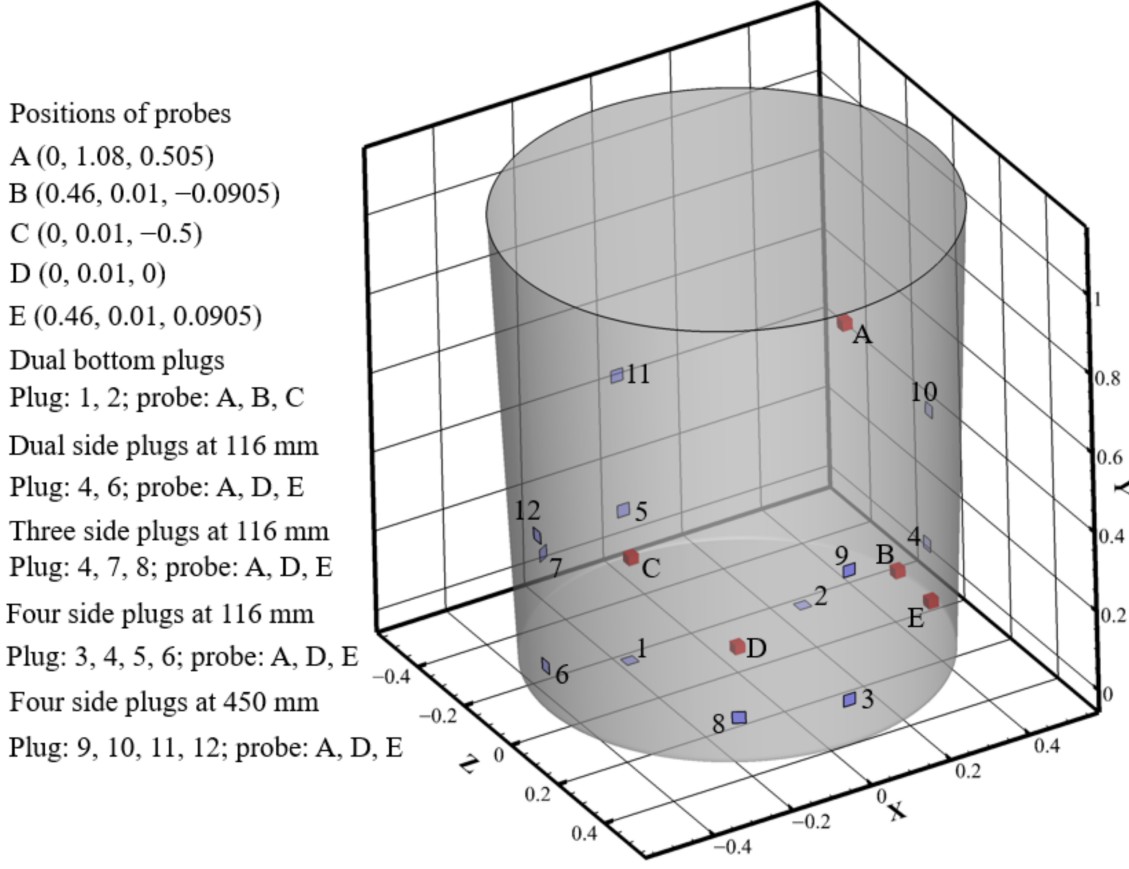

**Figure 2.** Three-dimensional coordinate system of the model ladle.

First, the gas (nitrogen) was injected through the plugs until a quasi-steady-state flow pattern was reached, the tracer (500 mL saturated KCl solution or 50 mL color dye) was injected into water through a funnel at the center of the water surface. The output signals of electrical conductivity probes were recorded for every 0.1 s by a personal computer combined with a data acquisition system. The mixing phenomenon of color dye was recorded by a video camera (Sony, Shanghai, China). The mixing time was determined as the tracer concentration from modified conductivity curves, which was continuously within ±5% of a well-mixed bulk value [13]. Three measurements were made for each operating condition, and an average mixing time was thereby determined. Figure 3a shows one conductivity curve, which was measured by a probe during gas blowing. $C_0$ is the conductivity of water. $C_e$ is the conductivity of water and the tracer. In order to eliminate the interference of water conductivity, the modified conductivity curves, i.e., conductivities of dual bottom plugs under the flow rate of 1.2 m$^3$/h minus the water conductivity are shown in Figure 3b. Figure 3b shows that the conductivity curves of the probe at different positions are different. The sequence of the tracer achievement is position A, position B, and position C in the ladle with dual bottom plugs. The measured mixing times of position B and C are longer than that of position C. This condition was influenced by the fluid flow path and the stagnant region and will be discussed in Section 5.

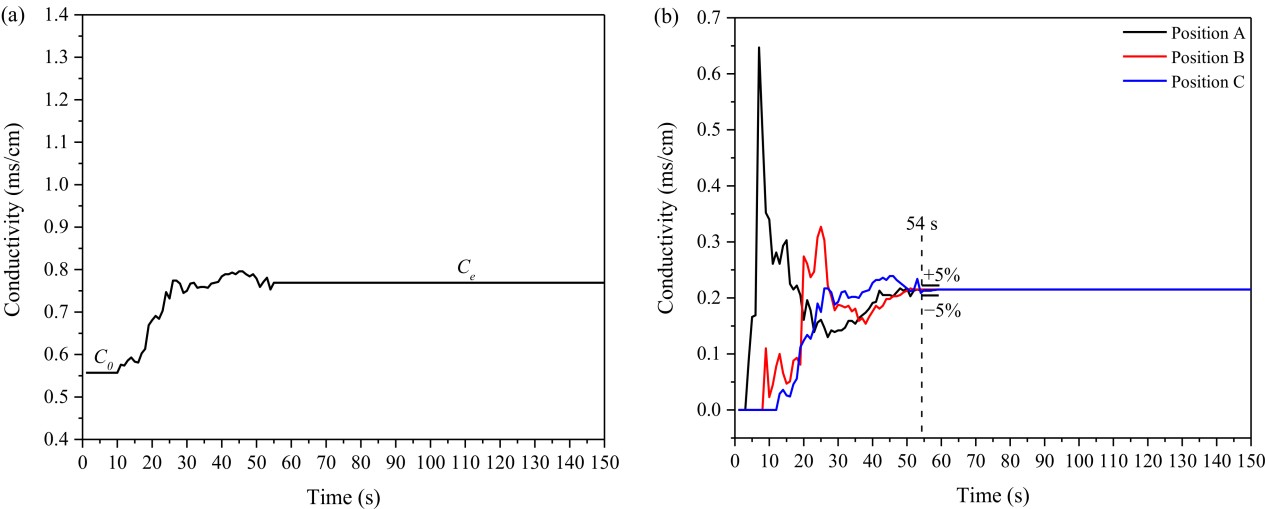

**Figure 3.** Conductivity curve of (**a**) water and tracer and (**b**) conductivity curves of tracer with dual bottom plugs under the flow rate of 1.2 m$^3$/h.

## 3. Numerical Modeling

The numerical modeling involves three parts: the water flow model, bubble transport model, and species transport model.

### 3.1. Assumptions

In order to simplify the numerical modeling, the present work includes the following assumptions:

1. The water is a homogeneous, incompressible Newtonian fluid.
2. The discrete bubbles generally take the shape of a sphere. The breakage of bubbles is neglected, and the bubbles have no expansion.
3. The air above water is ignored.

The present work considers the bubble coalescence but ignores the bubble breakage, because it is found that, in the present conditions, bubbles are small and few bubbles break up when going up in the water model. The temperatures of water and nitrogen gas are almost the same. The diameter of a bubble at water surface is just added up to 1.035 times of origin value when the water pressure is considered using ideal gas law. Therefore, the expansion of bubbles is ignored. The assumption that the air above water is ignored was based on the previous studies [37,39]. Figure 4 shows the mixing times of the experiment, the simulation that ignores the air, and the simulation that considers the air of the model ladle with four side plugs at 116 mm under different flow rates. The mixing times of the experimental and simulated results all decrease as the flow rate increases. The difference ratios of simulated mixing time between the two models—ignores air and considers air—are less than 15%. A primary goal of the present paper is analyzing the effects of the uniformity of water velocity distribution and the stagnant region on the mixing time. It is difficult for the results of the simulation that considers the air above water to provide precise values of the stagnant region rate and the water velocity distribution. Therefore, the air above water is ignored.

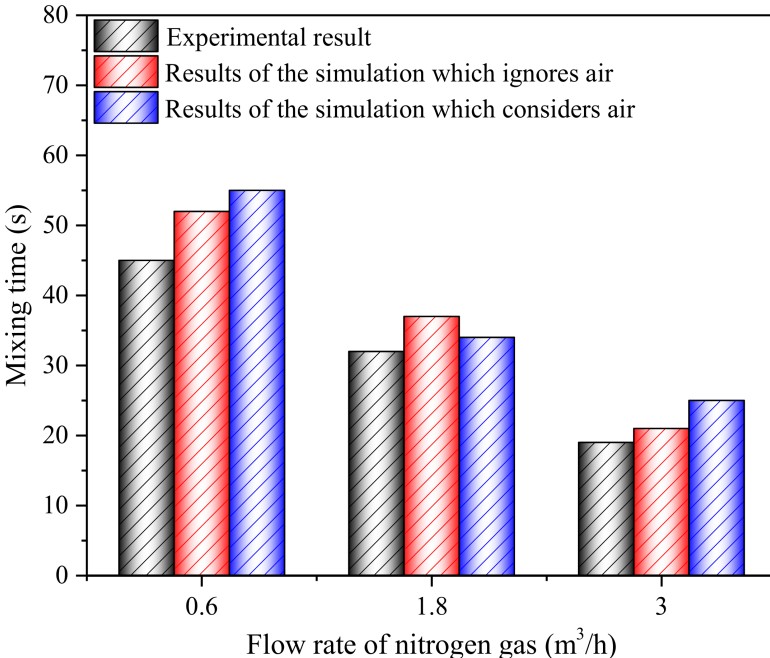

**Figure 4.** Mixing times of experiment, the simulation that ignored the air, and the simulation that considered the air of the model ladle with four side plugs at 116 mm under different flow rates.

*3.2. Fluid Flow Model*

Equation of continuity:

$$\nabla\left(\rho_l \vec{u}_l\right) = 0 \tag{3}$$

Momentum equation:

$$\frac{\partial \rho_l \vec{u}_l}{\partial t} + \nabla \cdot \left(\rho_l \vec{u}_l \vec{u}_l\right) = \nabla \cdot \left(\mu_e \nabla \vec{u}_l\right) - \nabla P - \vec{S}_b \tag{4}$$

where $\vec{u}_l$ is the fluid velocity (m/s), $\rho_l$ is the water density (1000 kg/m³), $P$ is the pressure (Pa), $\mu_e$ is the effective turbulent viscosity (Pa·s), and $\vec{S}_b$ is the momentum transfer term exerted by the discrete bubbles (kg/(m²·s²)).

The standard two-equation $k - \varepsilon$ model is used to model turbulence, which solves two equations for the transport of turbulent kinetic energy and its dissipation rate to obtain the effective viscosity field:

$$\mu_e = \mu_l + \mu_t = \mu_l + \rho_l C_\mu \frac{k^2}{\varepsilon} \tag{5}$$

where $\mu_l$ is the water viscosity with a value 0.001 Pa·s [42].

Turbulent kinetic energy $k$ (m²/s²):

$$\frac{\partial(\rho_l k)}{\partial t} + \nabla \cdot \left(\rho_l \vec{u}_l k\right) = \nabla \cdot \left[\left(\mu_l + \frac{\mu_t}{\sigma_k}\right) \nabla k\right] + G - \rho_l \varepsilon \tag{6}$$

where $G$ is the generation of turbulence kinetic energy due to mean velocity gradients:

$$G = \mu_t \frac{\partial u_{l,j}}{\partial x_i}\left(\frac{\partial u_{l,j}}{\partial x_i} + \frac{\partial u_{l,i}}{\partial x_j}\right) \tag{7}$$

The rate of dissipation of turbulent kinetic energy, $\varepsilon$ (m$^2$/s$^3$):

$$\frac{\partial(\rho_l\varepsilon)}{\partial t} + \nabla \cdot \left(\rho_l\vec{u}_l\varepsilon\right) = \nabla \cdot \left[\left(\mu_l + \frac{\mu_t}{\sigma_\varepsilon}\right)\nabla\varepsilon\right] + \frac{k}{\varepsilon}(C_1G - C_2\rho_l\varepsilon) \tag{8}$$

where $C_1$, $C_2$, $C_\mu$, $\sigma_k$, and $\sigma_\varepsilon$ are the empirical constants, whose values are 1.38, 1.92, 0.09, 1.0, and 1.3, respectively [40].

### 3.3. Bubble Transport Model

The movement of the particles is governed by the particle force balance equation defined as follows:

$$\frac{\pi\rho_b d_b^3}{6}\frac{\partial\vec{u}_b}{\partial t} = \vec{F_D} + \vec{F_p} + \vec{F_b} + \vec{F_V} + \vec{F_l} \tag{9}$$

where $\rho_b$ is bubble density with a value 1.25 kg/m$^3$, $d_b$ is particle diameter (m), $\vec{u}_b$ is bubble velocity (m/s), $\vec{F_D}$ is particle drag force (N), $\vec{F_p}$ is pressure gradient force (N), $\vec{F_b}$ is buoyancy force (N), $\vec{F_V}$ is virtual mass force (N), $\vec{F_l}$ is lift force (N). Details of these forces can be seen in previous works [43,44]. In this model, the chaotic effect of turbulence in the water on the bubble trajectories is considered using the random walk model [40].

For the description of the coalescence process, O'Rourke's algorithm [45] is used. Bubbles are considered to coalescence or bounce by comparing the actual collision parameter and the critical offset, which is a function of the collisional Weber number and the relative radii of the collector ($r_1$) and the smaller ($r_2$) one [46,47]:

$$b_{cri} = (r_1 + r_2)\sqrt{\min\left(1, \frac{2.4f}{We}\right)} \tag{10}$$

where $f$ is a function of $r_1/r_2$ defined as follows:

$$f(r_1/r_2) = (r_1/r_2)^3 - 2.4(r_1/r_2)^2 + 2.7(r_1/r_2) \tag{11}$$

### 3.4. Species Transport Model

The equation used for species transport model is:

$$\frac{\partial\rho_A C_A}{\partial t} + \nabla \cdot \left(\rho_A\vec{u}_l C_A\right) = \nabla \cdot \left(\left(\rho_A D_A + \frac{\mu_t}{Sc_t}\right)\nabla \cdot C_A\right) \tag{12}$$

where $C_A$ is the mass fraction of the saturated KCl solution, $\rho_A$ the density of the saturated KCl solution with a value of 1174 kg/m$^3$, $D_A$ is the diffusion coefficient of the specie with a value of $1.841 \times 10^{-5}$ cm$^2$/s [48], $Sc_t$ is the turbulent Schmidt number, which is set to 0.7 [40].

### 3.5. Boundary Condition and Numerical Details

Structured hexahedral grids were used in this work. Because the discretization error may be influenced by the calculation grid size, the mesh sensitivity should be firstly checked with different grids. Table 3 shows that three grid numbers of A, B, and C for each blowing mode were used to construct the calculation systems. The grid number of B was based on a previous work [49] and was set to around 500,000 elements. The comparisons of water average velocity and mixing time among three grid numbers of A, B, and C of different blowing modes under the flow rate of 1.2 m$^3$/h are shown in Table 4. The differences of calculation results of different blowing modes with the grid of number B and number C were all within 5% for the mixing time and average velocity of water. The highest difference ratios of A and C are more than 10%. Therefore, a domain with the grid number B was used to compare the effects of different parameters on the calculation results for each blowing mode. The mesh system of the model ladle with four side plugs at

116 mm were determined as shown in Figure 5. The mesh qualities of different blowing modes are shown in Table 5. To achieve correct results near the wall, the wall distances of the first elements are shown in Table 5, which ensured that the dimensionless wall distance (y+) was 1.

**Table 3.** Three grid numbers for each blowing mode.

| Blowing Mode | Grid Number | | |
|---|---|---|---|
| | A | B | C |
| Dual bottom plugs | 264,652 | 527,886 | 1,056,644 |
| Dual side plugs at 116 mm | 272,764 | 544,784 | 1,081,806 |
| Three side plugs at 116 mm | 288,640 | 581,262 | 1,166,842 |
| Four side plugs at 116 mm | 305,282 | 608,842 | 1,216,420 |
| Four side plugs at 450 mm | 295,840 | 592,880 | 1,182,620 |

**Table 4.** Comparisons of water average velocity and mixing time.

| Blowing Mode | Mixing Time (s) | | | | | | Average Velocity of Water (m/s) | | | | | |
|---|---|---|---|---|---|---|---|---|---|---|---|---|
| | A | | B | | C | | A | | B | | C | |
| | Time | Error | Time | Error | Time | | Velocity | Error | Velocity | Error | Velocity | |
| Dual bottom | 54 | 11.5% | 59 | 3.3% | 61 | | 0.04675 | 7.1% | 0.04486 | 2.7% | 0.04366 | |
| Dual side at 116 mm | 80 | 11.1% | 87 | 3.3% | 90 | | 0.04866 | 6.9% | 0.04624 | 1.5% | 0.04554 | |
| Three side at 116 mm | 44 | 6.4% | 46 | 2.1% | 47 | | 0.04996 | 4.4% | 0.04876 | 1.9% | 0.04785 | |
| Four side at 116 mm | 46 | 6.1% | 47 | 4.1% | 49 | | 0.05379 | 4.9% | 0.05243 | 2.2% | 0.05129 | |
| Four side at 450 mm | 59 | 10.6% | 63 | 4.5% | 66 | | 0.03995 | 8.1% | 0.03743 | 1.3% | 0.03696 | |

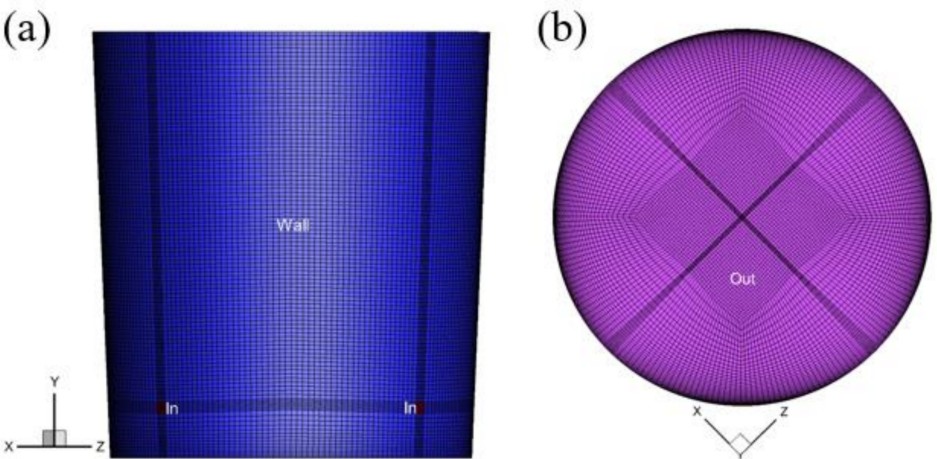

**Figure 5.** Mesh system of the model ladle: (**a**) front view; (**b**) top view.

**Table 5.** Mesh qualities.

| Blowing Mode | Aspect Ratio | Determinant | Wall Distance |
|---|---|---|---|
| | Maximum | Minimum | First Element |
| Dual bottom plugs | 11.675 | 0.782 | $1.52 \times 10^{-5}$ m |
| Dual side plugs at 116 mm | 11.786 | 0.794 | $0.92 \times 10^{-5}$ m |
| Three side plugs at 116 mm | 11.352 | 0.765 | $1.11 \times 10^{-5}$ m |
| Four side plugs at 116 mm | 11.243 | 0.802 | $1.32 \times 10^{-5}$ m |
| Four side plugs at 450 mm | 11.348 | 0.793 | $1.23 \times 10^{-5}$ m |

The simulation of gas agitation in a one-third scale model ladle was established using the open-source software OpenFOAM (the OpenFOAM foundation ltd, London, England), which has been used in previous studies [50]. In this modeling, the air above water is ignored, and the interface between the air and water is set as the out. For the simulation of fluid flow, no-slip boundary condition was used at the bottom wall, side wall, and the out, with standard "wall functions" in order to capture the steep gradients with reasonable accuracy on a coarse grid. The nitrogen bubbles were assumed to escape at the out and be reflected at the bottom wall and side wall. For the species transport for the tracer tracking, the zero flux boundary condition was used at walls and the out. The calculations were carried out by the transient-pressure-based solver. The bounded second-order implicit transient formulation and the bounded central-differencing scheme for momentum were used. The conservation equations were discretized using the control volume technique and the PISO (pressure-implicit with splitting of operators) scheme was used for the pressure–velocity coupling. The time step was set to 0.001 s. The CFL (CFL condition: an numerical method can be convergent only if its numerical domain of dependence contains the true domain of dependence of the PDE, at least in the limit as dt and dx go to zero. CFL is named by the three researchers' names of Courant, Friedrichs and Lewy ) number should be less than 2. The convergence criteria are set to $10^{-5}$ for the residuals of all dependent variables.

The initial bubble diameter was determined according to the experimental measurement. The initial bubble was assumed to be 1 mm as the diameter of the smallest bubbles observed in the bath is about 1 mm. The injection velocity of bubbles and the number of injected bubbles per unit time were calculated in Equations (13) and (14), respectively.

$$u_{b,0} = \frac{60 \times Q_{nitrogen}}{1000 \times (6S)} \tag{13}$$

$$n_b = \frac{60 \times Q_{nitrogen} \times 6}{1000 \times \left(\pi d_{b,0}^3\right)} \tag{14}$$

where $u_{b,0}$ is the injection velocity of bubbles (m/s), $d_{b,0}$ is the diameter of the initial bubble (0.001 m), and $n_b$ is the number of injected bubbles per unit time ($s^{-1}$).

## 4. Verification for Numerical Modeling

Table 6 shows the comparison of the mixing time for saturated KCl solution between the physical and numerical modelings under different blowing modes. The prediction errors for mixing time of the numerical modeling are less than 15%. The mixing times of numerical modeling agree well with the results of the physical modeling.

**Table 6.** Comparison of the mixing time for saturated KCl solution between the physical and numerical modelings.

| Blowing Mode | Mixing Time | | | | | |
| --- | --- | --- | --- | --- | --- | --- |
| | 1.2 m³/h | | | 2.4 m³/h | | |
| | Physical | Numerical | Error | Physical | Numerical | Error |
| Dual bottom plugs | 55 s | 59 s | 7.27% | 42 s | 40 s | 4.76% |
| Dual side plugs at 116 mm | 79 s | 87 s | 10.12% | 43 s | 49 s | 13.95% |
| Three side plugs at 116 mm | 47 s | 46 s | 2.13% | 32 s | 35 s | 9.38% |
| Four side plugs at 116 mm | 42 s | 47 s | 11.90% | 27 s | 29 s | 7.41% |
| Four side plugs at 450 mm | 66 s | 63 s | 4.55 % | 45 s | 51 s | 13.33% |

Figure 6 shows the velocity fields of the PIV (particle image velocimetry) result for case V of the previous study of Gajjar et al. [51], and the result of a simulation based on the dimensional parameters, blowing parameters of the case V in Gajjar's study [40], and the numerical modeling in Section 3. The comparison of Figure 6a,b shows that the velocity

magnitudes of water in the model ladle; except the plume zone of the PIV, result are almost the same as that of the simulated result. In Gajjar's work [51], since the bubble plume is very dense, the void fraction in this region is very high. Consequently, fewer tracers could be detected for the evaluation of the flow field. Thus, the evaluation becomes vague in the plume region (r = 0.35 m). Furthermore, the acrylic glass model is produced of two parts. Thus, an adhesive seam is present (r = 0.45 m), where the image is highly distorted. Therefore, the velocity magnitude of water in plume zone of the PIV result is lower than that of the simulated result. The flow paths of water in the PIV and simulated results are almost the same. In the PIV and simulated results, the bulk liquid both moved upward in the plume region. Reaching the free surface, the flow is redirected radially away from the plume. At the upper left corner, a circulation can be observed in all measurements, transferring some of the liquid's momentum into the deeper regions of the ladle and other toward the plume region or the free surface. The momentum transfer into the deeper regions, combined with the sucking of liquid into the plume causes the characteristic circulating flow. This is in accordance with Joo [52] and Liu [40], who reported the presence of a circulating flow in the ladle with one bottom plug in earlier studies. Therefore, the reliability of the numerical modeling could be verified by the above comparisons.

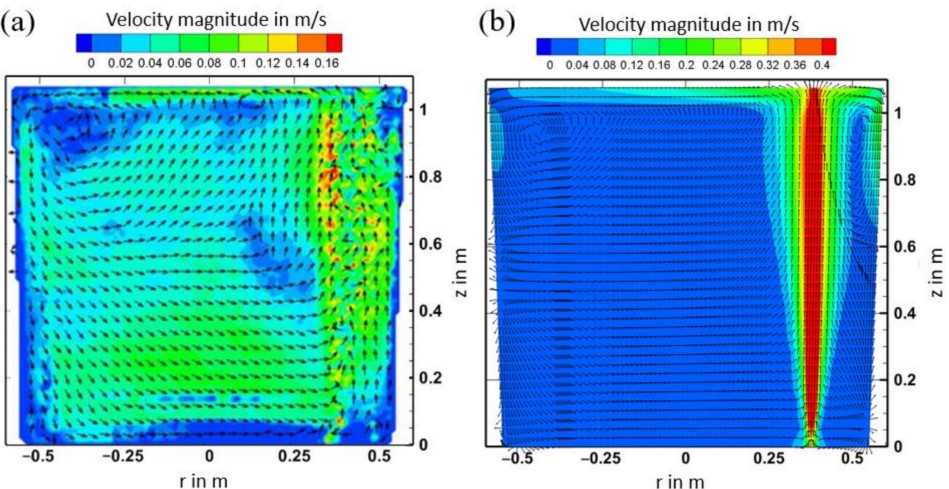

**Figure 6.** Velocity fields of particle image velocimetry (PIV) and simulated results: (**a**) PIV result [51]; (**b**) simulated result.

## 5. Results and Discussion

### 5.1. Comparison for Dual Bottom Plugs and Dual Side Plugs

A zone with water velocity that is lower than the average velocity of water has a poor fluidity, and the lower water velocity corresponds to the poorer fluidity. The zone with poor fluidity may be detrimental to tracer homogenization. The influence of the distribution of the velocity that is lower than the average velocity on the mixing condition is studied in this paper. The water velocity from 0 to the average velocity is divided into 10 equal groups. The volume fractions for the 10 groups of dual side plugs at 116 mm under the flow rates of 0.6 and 2.4 m$^3$/h are shown in Figure 7. The volume fraction of the low velocity under the flow rate of 0.6 m$^3$/h is higher than that of 2.4m$^3$/h. The distribution of the velocity that is lower than the average velocity under the flow rate of 2.4 m$^3$/h is more uniform than that of 0.6 m$^3$/h.

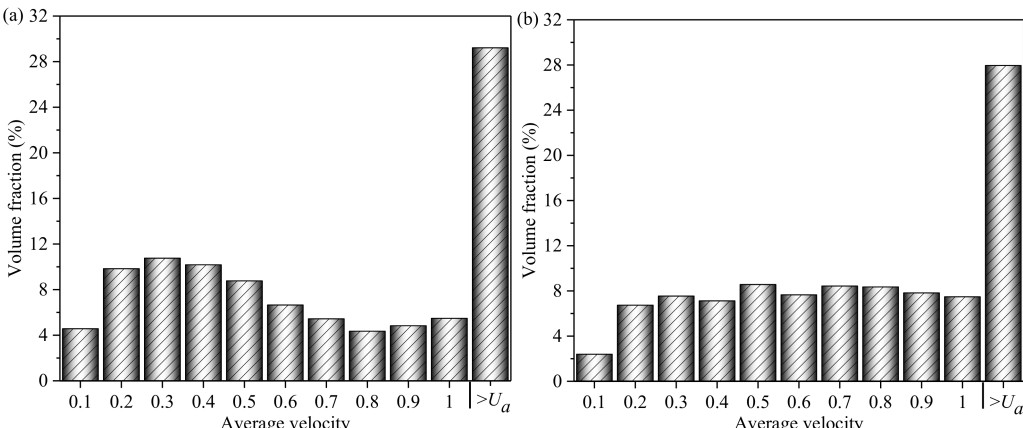

**Figure 7.** Distribution of the velocity that is lower than the average velocity under the flow rates of 0.6 and 2.4 m$^3$/h: (**a**) 0.6 m$^3$/h; (**b**) 2.4 m$^3$/h.

Variance ($D(U)$) in Equation (15) is often used to represent the dispersion of a discrete distribution. $U_i$ and $p_i$ are the average velocity and volume fraction of the group $i$, respectively. As the value of ($U_i - U_a$) in Equation (15) was easily influenced by the value of average velocity ($U_a$) of all water in the model ladle. $D(U)$ may be inaccurate to compare the uniformity of velocity distributions with the different $U_a$. Therefore, a modified variance ($D_m(U)$), which could clear up the disturbance of $U_a$ is shown in Equation (16). The velocity distribution of the case with a small value of the modified variance was more uniform than that of the case with a large value of the modified variance.

$$D(U) = \sum_{i=1}^{10} (U_i - U_a)^2 p_i \tag{15}$$

$$D_m(U) = \sum_{i=1}^{10} \left( \frac{U_i - U_a}{U_a} \right)^2 p_i \tag{16}$$

Figure 8a shows the mixing times of dual bottom plugs and dual side plugs in physical modeling. Figure 8b shows the water flow parameters of the two blowing modes after blowing 60 s in numerical modeling. In this paper, the water flow parameters in numerical modeling are all after blowing 60 s. During the gas agitation, the water temperature and the vertical height of whole water in the model are almost unchanged. Only the water velocity changes. Therefore, the water average velocity represents the agitation power of gas. The water average velocity of dual bottom plugs was lower than that of dual side plugs under the same flow rate of nitrogen gas. However, when the flow rate of nitrogen gas was only higher than 2.4 m$^3$/h, the mixing time of dual bottom plugs was longer than that of dual side plugs at 116 mm. Therefore, the mixing time was not influenced by the agitation power alone.

The stagnant region is defined as the region with velocity that is lower than 1/10 the average velocity of water in the model ladle. $V$ is the volume fraction of stagnant region. Figure 9a,b shows water flow paths of dual bottom plugs under the flow rates of 0.6 and 3.0 m$^3$/h, respectively. As the flow rate of gas increases, the water velocity increases, but the water flow paths have little change. The water flowing which is driven by bubbles is mainly along the vertical direction in the ladle with bottom blowing. The water at the bottom edge of the ladle is difficult to be driven to flow. Therefore, the fluidity of water at the bottom edge of the model ladle is poor. From Figure 9c,d, as the flow rate increases, the $V$ decreases slightly, but the positions of the stagnant regions have little change. There is still a volume of stagnant region at the bottom of the model ladle under the flow rate of 3.0 m$^3$/h.

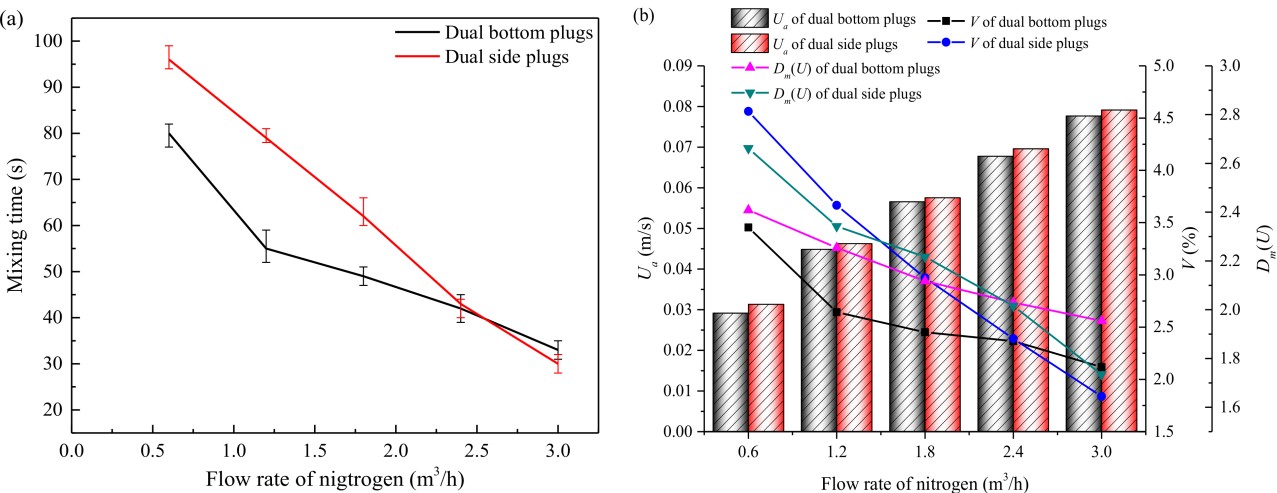

**Figure 8.** Mixing time and flow parameters of water in model ladle under two blowing modes: (**a**) mixing time; (**b**) flow parameters of water.

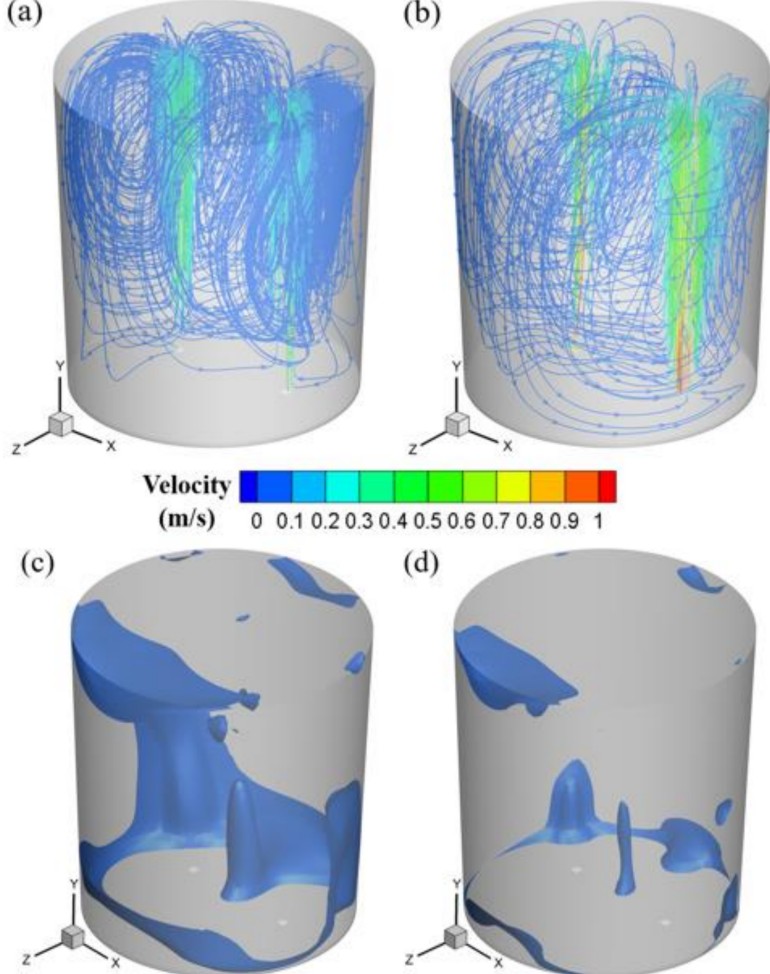

**Figure 9.** Flow parameters of water in model ladle with dual bottom plugs under different flow rates: (**a**) streamline of 0.6 m³/h; (**b**) streamline of 3.0 m³/h; (**c**) stagnant region of 0.6 m³/h; (**d**) stagnant region of 3.0 m³/h.

Figure 10a,b shows the water flow paths of dual side plugs under the flow rates of 0.6 and 3.0 m$^3$/h, respectively. As the flow rate of gas increases, the injecting velocity of gas is improved. The increased inertia force of gas bubbles enhances the horizontal flow of water at the bottom. Therefore, the fluidity of water at the bottom of the ladle was improved. From Figure 10c,d, as the flow rate of gas increases, the $V$ also decreases, and the stagnant region at the bottom of ladle reduces evidently.

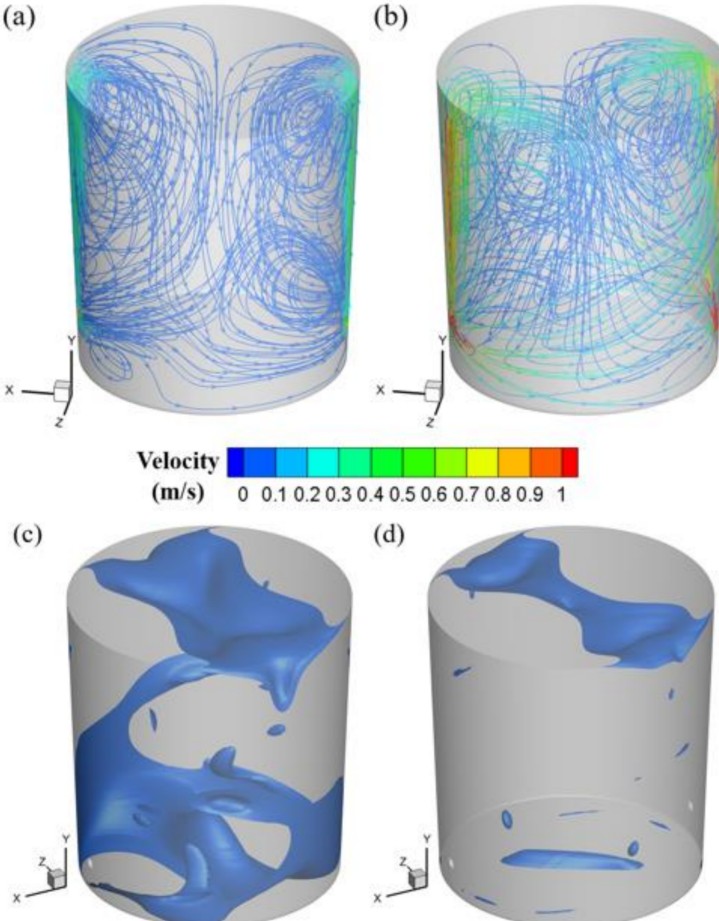

**Figure 10.** Flow parameters of water in model ladle with dual side plugs at 116 mm under different flow rates: (**a**) streamline of 0.6 m$^3$/h; (**b**) streamline of 3.0 m$^3$/h; (**c**) stagnant region of 0.6 m$^3$/h; (**d**) stagnant region of 3.0 m$^3$/h.

### 5.2. Influence of Side Plug Number

Figure 11a shows the mixing times with different numbers of side plugs at 116 mm in physical modeling. More side plugs and high flow rate of nitrogen gas both contribute to the short mixing time. Figure 11b shows that more side plugs cause the larger average velocity of water, lower $V$ and $D_m(U)$. Therefore, more side plugs led to a short mixing time. The average velocity of water and water velocity distribution of three side plugs and four side plugs are larger and more uniform than that of dual bottom plugs. Therefore, the mixing times of three side plugs and four side plugs are shorter than that of dual bottom plugs.

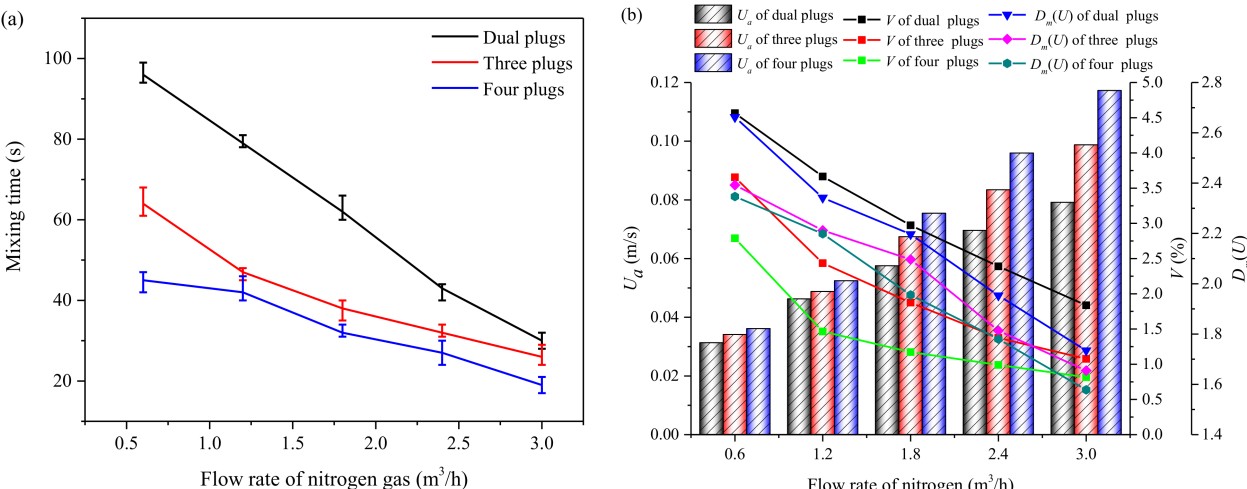

**Figure 11.** Mixing time and flow parameters of water in model ladle with different number of side plugs at 116 mm: (**a**) mixing time; (**b**) flow parameters.

Figure 12 shows the flow paths with different numbers of side plugs under the flow rate of 1.8 m³/h at 116 mm after blowing 60 s. The water velocity of horizontal flow is reduced with the increased distance from the plug. Increased number of side plugs cut down the distance between two neighboring plugs. Therefore, increased number of side plugs improves the water velocity of horizontal flow at the bottom of the ladle in red circles of Figure 12.

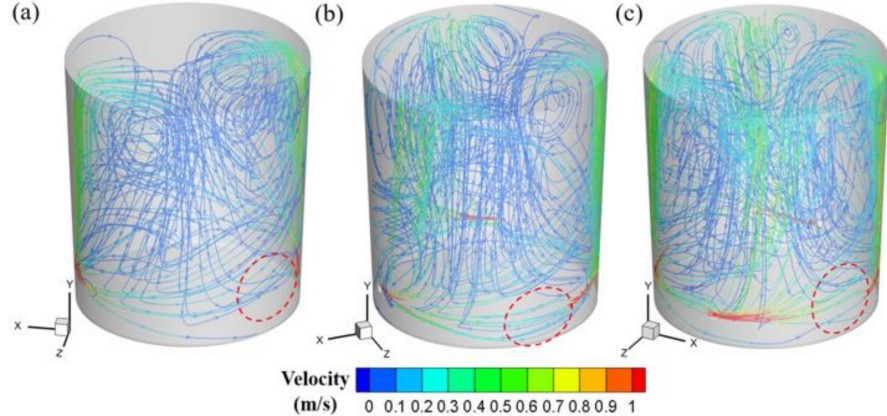

**Figure 12.** Streamlines with different numbers of side plugs at 116 mm: (**a**) dual plugs; (**b**) three plugs; (**c**) four side plugs.

Figure 13 shows the stagnant regions with different number of side plugs at 116 mm under the flow rate of 1.8 m³/h. The larger horizontal velocity, which is caused by more side plugs, led to the smaller *V* at the ladle bottom. Because of the uniform distribution of side plugs, the stagnant regions at ladle top in Figure 13 appear in the confluence area, which is caused by the collision among the liquid streams in different directions in Figure 12. The horizontal-rotation flow of water could alleviate the stagnant region, which is caused by confluence. Figure 14 shows the velocity profiles in cross section at 1000 mm of model ladle bottom with dual and four side plugs under the flow rate of 1.8 m³/h, respectively. The horizontal-rotation flow of water close to the wall of four side plugs in Figure 14b causes no formation of a stagnant region of water close to the wall in Figure 13c. The collision among the liquid streams at ladle top of four side plugs in Figure 12c could not interfere with the horizontal-rotation flow of water close to the wall in red circles of Figure 14b forming. Therefore, there are no stagnant region of water close to the wall at the

ladle top forming. The collision among the liquid streams at ladle top of dual side plugs in Figure 12c interfered with the horizontal-rotation flow of water close to the wall in red circles of Figure 14a forming. The stagnant region of water close to the wall at the ladle top formed in Figure 13a.

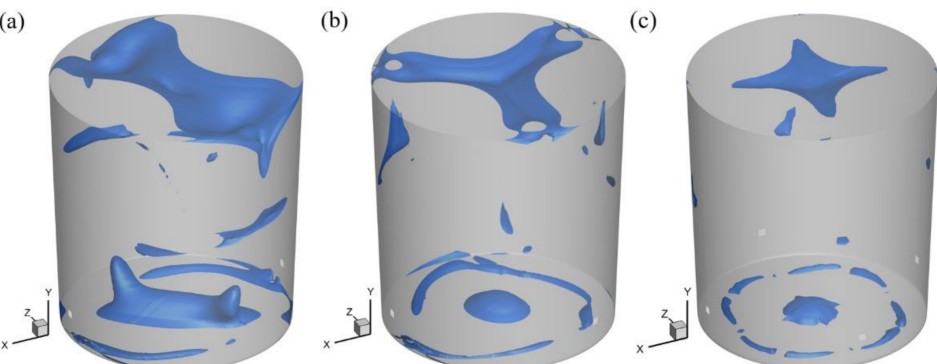

**Figure 13.** Stagnant regions with different number of side plugs at 116 mm: (**a**) dual side plugs; (**b**) three plugs; (**c**) four side plugs.

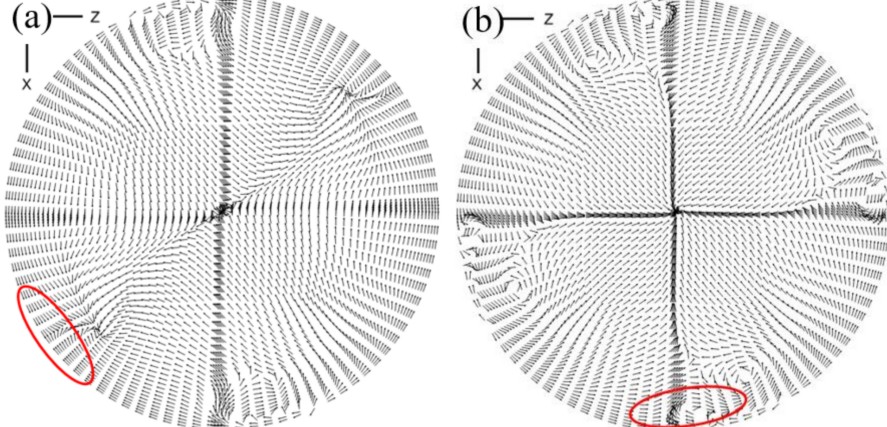

**Figure 14.** Velocity profiles in cross section at 1000 mm of model ladle with different number of side plug: (**a**) dual side plugs; (**b**) four side plugs.

### 5.3. Influence of Side Plug Height

Figure 15a shows the mixing times of four side plugs at 116 and 450 mm in physical modeling. The high flow rate of side plugs at two heights both contribute to the short mixing time. However, the mixing time of 116 mm is much longer than that of 450 mm under the same flow rate. Due to buoyancy and inertial force, the gas bubbles that were entering into the water in the model ladle move along obliquely upward. Compared with the side plugs at 116 mm in Figure 16a, limited water below the plug height could be driven to flow by bubbles from side blowing at 450 mm in Figure 16b. From Figure 16c,d, the stagnant region rate at ladle bottom of the side plugs at 450 mm is much larger than that for the side plugs at 116 mm. Nevertheless, the stagnant region rate at the top of the four side plugs at 450 mm is little smaller than that of the four side plugs at 116 mm. That is because, the water at the top of model ladle is driven by the bubbles of side plugs at 450 mm to horizontal flow more strongly.

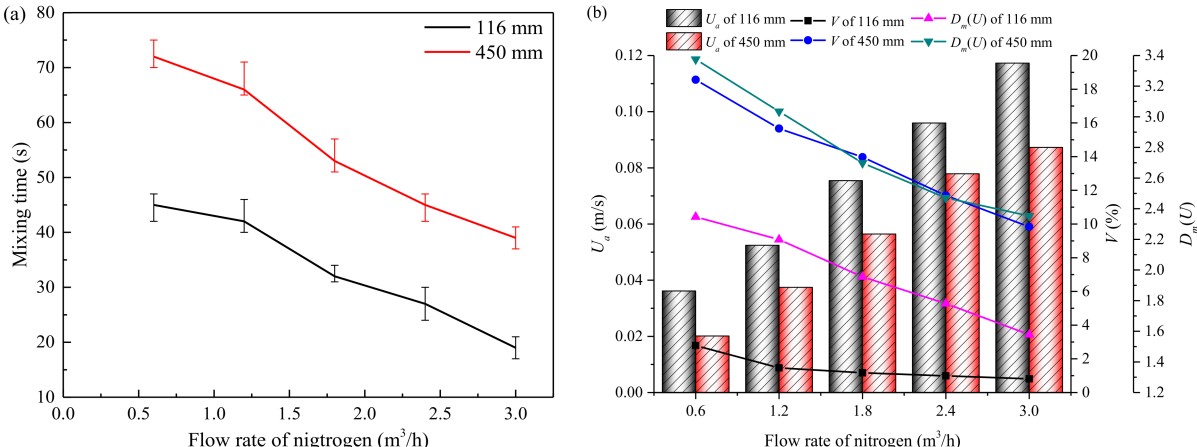

**Figure 15.** Mixing time and flow parameters of water in model ladle with four side plugs at different heights: (**a**) mixing time; (**b**) flow parameters of water.

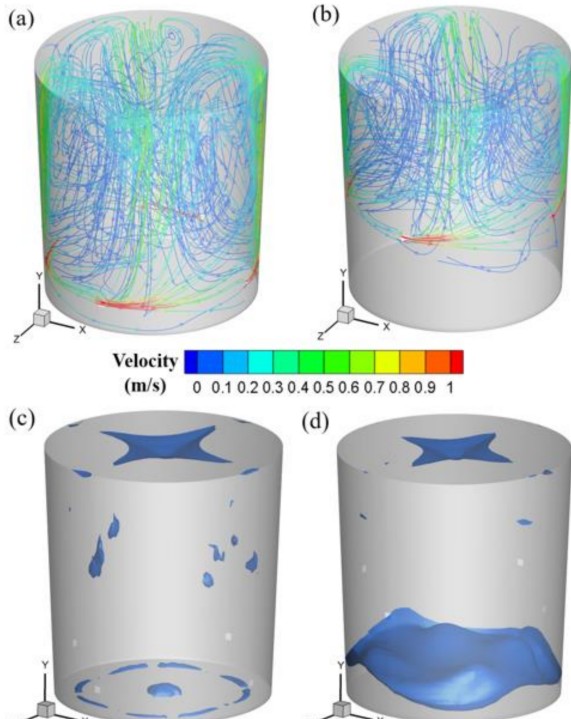

**Figure 16.** Streamlines and stagnant regions of water in model ladle with four side plugs at different heights under the flow rate of 1.8 m³/h after blowing 60 s: (**a**) streamline at 116 mm; (**b**) streamline at 450 mm; (**c**) stagnant at 116 mm; (**d**) stagnant at 450 mm.

Figure 15b shows the flow parameters of water with four side plugs at 116 and 450 mm under the flow rate of 1.8 m³/h in numerical modeling. Because the distance of buoyancy working of the four side plugs at 450 mm is shorter than that of the four side plugs at 116 mm. The average velocity of water of 450 mm is lower than that of 116 mm, and the $D_m(U)$ and $V$ of 450 mm are larger than that of 116 mm. Therefore, the mixing time of 116 mm is much lower than that of 450 mm under the same flow rate of nitrogen gas. In spite of this, the water average velocities of 450 mm under the flow rates of 1.2 and 1.8 m³/h are both larger than that of 116 mm under the flow rate of 0.6 m³/h. The mixing times of 450 mm under the flow rates of 1.2 and 1.8 m³/h are both shorter than that of 116 mm under the flow rate of 0.6 m³/h. That is because, $D_m(U)$ and $V$ of 450 mm under the flow rates of 1.2 and 1.8 m³/h were both larger than that of 116 mm under the flow rate

of 0.6 m³/h. The mixing time was not only influenced by the agitation power but also by the uniformity of velocity distribution.

### 5.4. Diffusion Paths of Bottom Blowing and Side Blowing

Convective flow and turbulent diffusion were the two primary transport mechanisms of the mixing phenomenon [30]. The fluid flow plays a major role in the tracer diffusion in model ladle. Therefore, the diffusion paths of color dye and KCl solution are almost the same under the same blowing mode and flow rate. Table 7 shows that the diffusion paths of color dye in physical modeling and KCl solution in numerical modeling are almost the same under the same blowing mode. The tracers diffuse down along the wall of ladle to the bottom and finally reach the bottom edge that is close to plugs with the force of liquid field in bottom blowing. Figures 12c and 14b show that side injecting gas spirals the water that is close to the side wall of the model ladle up, after the water is spiraled to the liquid surface, the water flows to the center and flows down along the center line to the bottom, and then reaches the bottom edge, the water is finally spiraled by the gas again. The tracers diffuse down along the center line to the bottom and finally reach the bottom edge in side blowing. The red circles correspond to the regions where the tracer finally arrives. The diffusion paths of the tracer of bottom blowing and side blowing in Table 7 are almost the same as the flow paths of water in Figures 9b and 12c, respectively. The regions where the tracer finally arrives for color dye in experiment and KCl solution in simulation are also the same as the parts of the stagnant regions of Figures 9 and 13. Therefore, the mixing phenomenon is influenced by the stagnant region and by the distribution uniformity of water velocity.

**Table 7.** Diffusion paths of KCl solution of bottom plugs and four side plugs at 116 mm in physical modeling and numerical modeling under the flow rate of 2.4 m³/h.

| Modes | | Early Stage | Middle Stage | Late Stage |
|---|---|---|---|---|
| Bottom blowing | Physical modeling |  |  |  |
| | Numerical modeling |  |  |  |
| Side blowing | Physical modeling |  |  |  |
| | Numerical modeling |  |  |  |

## 6. Conclusions

In this study, the fluid flowing and mixing condition in a model ladle with side blowing were investigated by physical and numerical modelings together. The following conduction were drawn:

1.  The reliability of numerical modeling could be verified by the same mixing times for KCl solution of experimental and simulated results, the agreement of velocity field between simulated result and the PIV result in Gajjar's work [51], and the same diffusion paths for color dye of experimental result and KCl solution of simulated result.

2.  The mixing time is not only influenced by the agitation power but also by the uniformity of fluid velocity distribution. Higher agitation power, more uniform velocity distribution of fluid, and lower stagnant region rate led to the shorter mixing time.

3.　The horizontal flow of water at the bottom was driven by the inertia force of gas bubbles injected into the water through the side plugs. The more numbers of side plugs and higher flow rate of gas both enhanced the horizontal flow. The effect of the inertia force of gas bubbles on the flow field of water should not be ignored.

4.　Compared with the bottom blowing, the shorter mixing time, larger average fluid velocity, and more uniform velocity distribution for four side blowing close to bottom could shorten the time of alloy homogenization and may contribute to the inclusion removal and desulfurization reaction. In future, the inclusion removal and desulfurization reaction in the ladle with side plugs will be studied.

**Author Contributions:** Conceptualization, R.C. and J.Z.; methodology, R.C.; software, R.C. and L.Z.; validation, R.C. and L.Z.; formal analysis, R.C. and J.Z.; investigation, Y.Y.; resources, Y.Y.; data curation, R.C. and J.Z.; writing—original draft preparation, R.C. and L.Z.; writing—review and editing, R.C. and J.Z.; visualization, Y.Y. and R.C.; supervision, Y.Y.; project administration, R.C. and J.Z.; funding acquisition, J.Z. All authors have read and agreed to the published version of the manuscript.

**Funding:** This research was funded by the National Natural Science Foundation of China, grant number 51834002.

**Data Availability Statement:** Not applicable.

**Conflicts of Interest:** The authors declare no conflict of interest.

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
