# Peer review of "Effect of Side Blowing on Fluid Flow and Mixing Phenomenon in Gas-Stirred Ladle"

_metals, doi:10.3390/met11020369_

Round 1

Reviewer 1 Report

The presented work investigates the effect of side blowing on mixing time and fluid motion in gas-agitated ladles via CFD and tracer path measurements.

The aim and goals of the study are quite clearly formulated and fit the journals topic well.

Nevertheless, the manuscript needs major revisions.

Further recommendations are given hereinafter:

Overall comments:

  • I don't consider the term “stirring” to be very accurate in the framework used. According to my understanding, stirring includes a mechanical stirring organ, which is missing here, however. “Gas agitation” or “gas mixing” seems more appropriate to me.
  • Please make sure your manuscript is spell-checked:
  • Some minor mistakes occur.
  • Sometimes grammar is not exact.
  • I miss the discussion about the comparability of the substance data from experiment and modeling with the real application.
  • Conductivity measurements and the positions of the measuring device should be discussed. Different positions will provide different curves.

Chapter 1:

  • In this context, Chapter 1 lacks some relevant literature work on the various techniques and topics addressed (PIV, Mixing time, CFD). E.g. following literature - as many others – could improve the experimental and simulative approach and could be discussed:
    • E. Jardón-Pérez, A. M. Amaro-Villeda, C. Gonzalez-Rivera, G. Trapaga, A. Conejo, M. A. Ramírez-Argáez, Metallurgical and Materials Transactions, July 2019, 50(5), DOI: 10.1007/s11663-019-01631-y.
    • Basheer, S. Pushpavanam, Chemical Engineering Journal, April 2012187:261–274, DOI: 10.1016/j.cej.2012.01.078.
    • Stefan, H. J. Schultz, in OpenFOAM (Eds: J. No´brega, H. Jasak), Springer, Cham 2019, 509 – 520. DOI: https://doi.org/10.1007/978-3-319-60846-4_36.
    • -S. Chae, B.-J. Chung, International Journal of Thermal Sciences, May 2021, 163(503):106776, DOI: 10.1016/j.ijthermalsci.2020.106776.
    • E. Jardón-Pérez, A. M. Amaro-Villeda, A. Conejo, C. Gonzalez-Rivera, M. A. Ramírez-Argáez, Materials and Manufacturing Processes, November 2017, DOI: 10.1080/10426914.2017.1401722.
    • Wolinski, M. Ulbricht, H. J. Schultz, July 2019, Chemie Ingenieur Technik, DOI: 10.1002/cite.201800099.
    • M. Amaro-Villeda, M. A. Ramírez-Argáez, A. Conejo, ISIJ International, January 2014, 54(1):1-8, DOI: 10.2355/isijinternational.54.1.
    • Jaehrling, H. J. Schultz, Flow fields in stirred vessels depending on different internal heat exchangers and vessel bottoms, Chem. Ing. Tech. 2019, 91 (9), 1281-1292, DOI: 10.1002/cite.201800101.
    • E. Jardón-Pérez, A. M. Amaro-Villeda, G. Trapaga-Martínez, C. Gonzalez-Rivera, M. A. Ramírez-Argáez, Metallurgical and Materials Transactions, August 2020, 51(3), DOI: 10.1007/s11663-020-01944-3
  • For example, the influence of the reactor bottom (flat/round) on energy consumption is not considered/discussed.
  • PIV is not discussed at all in chapter 1.
  • What about multi-stage impellers?
  • The paper does not consider the formation of spouts. This is not useful.
  • The turbulent kinetic energy is a useful measure for assessing turbulence and energy input. This is missing here.
  • Line 36: clarify what “PH tracer” exactly means, more details
  • Line 40: “percentage” and “%” doubled; clarify and explain this 95% homogenization degree as mixing criteria
  • Line 41/42: in the brackets switch R and L to fit the sequence

Chapter 2:

  • Are water/nitrogen suitable model for the investigation of flow for molten steel systems and argon? -> Discussion!
  • Line 97: the word “height” at the beginning doesn’t belong there
  • Figure 2: The coordinates in the legend don’t fit the marked points in the cylinder
  • Line 144: Three measurements for each experiment? Formulate clearer!
  • Figure 3: Lacking quality and resolution, hard to read
  • Figure 3: A curve that shows no fluctuations at all from about 55 s seems strange to me. Statistical fluctuations up and down to the usual extent would be expected. Why do these not occur here? Please check and clarify.

Chapter 3:

  • Line 207: 600,000 cells are few. In further studies, the number of cells should be increased. A sensitivity analysis should clarify whether the number of cells is sufficient or whether it still has a significant influence on the simulation. -> Include remark in outlook.

Chapter 4:

  • Line 235: Add the spatial resolution of the PIV with regards to the spatial resolution of the CFD simulations

Chapter 5:

  • Line 262: Replace “the” with “a”
  • Line 271: “uniform”
  • Figure 6: Correct the units in the description, m³ not m3
  • Line 289ff and Figure 7: The statement does not match the figure.
  • Line 323ff: same statement as above (line 289ff)
  • Figure 8, 9, 11, 15: I suggest to reduce the number of streamlines in (a) and (b) in order to make the figures more vivid. It is hard to follow the path of flow with that many streamlines.
  • Line 331: “more”
  • Line 343: delete “the”
  • Line 355: number and unit together in one line
  • Figure 12 (b): Copy-Button is visible on the screenshot
  • Line 372: “ladle”
  • Figure 15: In the description m³ instead of m3
  • Line 401: Chapter 5.4. starts with an incomplete sentence
  • Line 407: “that” is double
  • Table 4: In the title m³ instead of m3

Chapter 6:

  • No comments, ok

Reviewer 2 Report

Review of the manuscript entitled: Effect of Side Blowing on Fluid Flow and Mixing Phenomenon in Gas-stirred Ladle

The reviewed manuscript is very interesting. The simulated case is relatively difficult for modeling. On the other hand, the same situation concerns experimental measurements. However, the manuscript contains some lacks and elements that need to extend. A separate subject is preparing appropriate literature research in the field of CFD analysis of similar cases especially in the field of multiphase unsteady flow apache in the cylindrical tanks.
Below the main lacks and errors:
1. Some general extension of references about multiphase approaches (especially unsteady) should be presented. Generally, extended literature research in the above cases should be prepared. For example, check and follow:
https://doi.org/10.1016/j.jfoodeng.2019.109846
2. The simplifications should be discussed and judged.
3. The free surface (air ignored above fluid phase) simplification seems inadequate, especially when the shape of it depends on mixing efficiency.
4. Details of the mesh quality testing (morphology choice, number of elements counts testing, quality testing, quality parameters (y+?) ) should be provided. CFL should be discussed also.
5. Details on the criteria for the simulation model and its testing of convergence should be discussed.
6. Is it k-epsilon turbulence model really adequate for multiphase flow cases with air bubble mixing? What about coefficients that seem like default values?
7. What about your own PIV results compare? The referenced paper presents the bottom aeration only.
8. Tangentially air inflow generates a rotating flow. See the experimental pictures in table 4. What about it in the view of simulation results?
9. In the view of mechanical mixing efficiency, a power number is an important thing. The mixing efficiency for your case and discussion about it is necessary.
10. What about the scale-up? The model is in the 1:3 scale, so provide and discuss some proposals in this case. Dimensional similarity only seems inadequate.

Summary: I recommend some extensions and a major revision of the reviewed manuscript

Reviewer 3 Report

Dear Authors, Thanks for your submitted manuscript. It is really interesting. However, there are some points which is needed to be addressed clearly. Please find my comments below: 1- In abstract you need to say by your work how much you improved and what is your important findings. Make it more specific. Try to talk about some numbers and percentages! 2- What is the novelty of your work? We have plenty of such works quite similar to yours! They added tracers and also build real model and also the prototype. Could you mention your novelty at the last paragraph of your introduction? I am quite sure about the papers which are published in similar cases after 2014 up to now. They have replied many of your findings. 3- You have some equations and since you have incompressible flow you shouldn't write it in the continuity equation! Am I right? Or you wanted to mention the general form of it! 4- You need to provide a nomenclature which describes the dimension and the exact definition of each terms that you have used! 5- I didnt understand the physical model! Are you refer to your experimental test rig! Please write it as experimental test. It gives a better feeling! 6- Please add the reference and the complete citation requirements for Openfoam! 7- Indeed you tried hard and you have nice figures; however, you need to increase their quality anyway. 8- Your literature survey and also your reference list is very old. Please find more recent papers to cope with the real novelty of your work. 9- What about your simulation errors! Why you chose this range of meshes? Are you sure that you have done a mesh sensitivity test at least! 10- When you scaled your prototype, which dimensionless number you have focused to be sure that you are doing a proper approach!

Round 2

Reviewer 1 Report

Nice work, improvements are done well. Hope to read more of your research.

Only a small thing: Line 522: "may" is doubled.

Reviewer 2 Report

Re-review of the manuscript entitled: Effect of Side Blowing on Fluid Flow and Mixing Phenomenon in Gas-stirred Ladle
Like I found the manuscript is corrected well. The authors prepared good answers,  provide adequate corrections, and improve the manuscript. The re-reviewed paper has novelty aspects. The paper concern with the simulation of the multiphase flow in a gas-stirred ladle. It's an actual subject so the paper should interesting for the readers and should be citable also in the future. The paper has a practical aspect also. The authors provided wider literature research and expand a part of the manuscript concern with simulation methodology. This will be of benefit to the recognition of the manuscript among other authors which concerned with some similar cases. In summary: in view of responses and prepared corrections, I recommend accepting the manuscript in the present form.

Reviewer 3 Report

Dear Authors,

Many thanks for your all answers. As I can see you replied almost all of them with enough details and the paper now has a reasonable quality for publication.